# OpenReview forum: "Clipping Improves Adam-Norm and AdaGrad-Norm when the Noise Is Heavy-Tailed"
_ICML.cc/2025/Conference — ICML 2025 poster_

### Official Review · Reviewer_GE8p · 2025-03-07

**Overall Recommendation:** 4

**Summary:**

This paper studies high probability convergence rate of clip-AdamNorm and clip-AdaGradNorm under heavy-tailed noise. The authors show that Adam/AdaGrad fails to get convergence rate with $log(1/\delta)$ dependence if the noise is heavy-tailed. Then the authors show that gradient-clipping can fix this issue for AdamNorm and AdaGradNorm.

**Claims And Evidence:**

Yes

**Essential References Not Discussed:**

No

**Experimental Designs Or Analyses:**

Yes, in Section 4

**Methods And Evaluation Criteria:**

Yes

**Other Comments Or Suggestions:**

In Theorem 3.3, $M\rightarrow\Delta$

**Other Strengths And Weaknesses:**

This is a clearly theoreitical-solid paper, providing a complete proof of high probability convergence of Adam-type algorithms with clipping. The only concern is that the analysis heavily depends on the global adaptive stepsize instead of coordinate-wise scaling as in original Adam. Although the authors provide results for clip-Adam, it still dependends on $\beta_1=0$. This is not aligned with practice. The only paper I know that analyzes clip-Adam without sub-gaussian noise assumption is [1], which has similar assumptions with this paper but in distributed setting. This might be helpful for the authors to do future research.

[1] Cheng, Ziheng, and Margalit Glasgow. "Convergence of Distributed Adaptive Optimization with Local Updates." arXiv preprint arXiv:2409.13155 (2024).

**Questions For Authors:**

see strength and weakness part

**Relation To Broader Scientific Literature:**

It is generally related to theoretical understanding of popular optimization algorithm in machine learning.

**Theoretical Claims:**

No

---

> ### Author Rebuttal · Authors · 2025-04-01
>
> We thank the reviewer for the very positive evaluation of our paper and constructive feedback.
>
> >1:**Analysis of Clip-Adam with $\beta_1 > 0$**
>
> A: We revisited our proof and generalized it to the case of $\beta_1 > 0$. The sketch of the proof with the key derivation is provided below. We promise to include the complete proof in the final version.
>
> The main idea remains the same: we prove the descent lemma for the coordinate-wise approach as for the nonconvex case in scalar-wise form (Lemma C.3) and the rest of the proof follows the induction-based arguments (similarly to Theorem C.5). Applying smoothness and rewriting everything in the coordinate-wise form, we get an analog of formula (29) for Clip-AdaGradD and Clip-AdamD:
>
> $$f(x_{t+1}) - f(x_t) \leq \sum\limits_{i=1}^d\left[-\gamma \frac{\nabla_{t, i} m_{t,i}}{b_{t,i}} + \frac{L\gamma^2}{2}\frac{m_{t,i}^2}{b_{t,i}^2}\right].$$
>
> The next step is to derive an analog of formula (30) from page 26. We have:
>
> \begin{align}
> 	-\nabla_{t,i} m_{t,i} &= -\beta_1 \nabla_{t,i} m_{t - 1,i} - (1 - \beta_1)\nabla_{t,i} g_{t,i} \\\\
> &= -\beta_1 (\nabla_{t,i} - \nabla_{t-1,i}) m_{t - 1,i} -\beta_1 \nabla_{t-1,i} m_{t - 1,i} - (1 - \beta_1)\nabla_{t,i} g_{t,i} \\\\
> &\leq \beta_1 |\nabla_{t,i} - \nabla_{t-1,i}| \cdot |m_{t - 1,i}| -\beta_1 \nabla_{t-1,i} m_{t - 1,i} - (1 - \beta_1)\nabla_{t,i}g_{t,i}.
> \end{align}
>
> Unrolling the recurrence and summing over $i$, we get
>
> $$ -\sum\limits_{i=1}^d \frac{-\nabla_{t,i} m_{t,i}}{b_{t,i}} \leq -(1 - \beta_1)\sum\limits_{k=0}^t \beta_1^{t-k}\sum\limits_{i=1}^d\frac{\nabla_{k,i}g_{k,i}}{b_{t,i}} + \sum\limits_{k=0}^{t-1} \beta_1^{t-k} \sum\limits_{i=1}^d\frac{ |\nabla_{k+1,i} - \nabla_{k,i}| \cdot |m_{k,i}|}{b_{t,i}}. $$
>
> Applying Cauchy-Schwarz inequality to the last term in the right-hand side and then using $L$-smoothness, we obtain
>
> \begin{align}
> \sum\limits_{k=0}^{t-1} \beta_1^{t-k} \sum\limits_{i=1}^d\frac{|\nabla_{k+1,i} - \nabla_{k,i}| \cdot |m_{k,i}|}{b_{t,i}} &\leq \sum\limits_{k=0}^{t-1} \beta_1^{t-k} \left(\sqrt{\sum\limits_{i=1}^d (\nabla_{k+1,i} - \nabla_{k,i})^2}\right)\left(\sqrt{\sum\limits_{i=1}^d\frac{m_{k,i}^2}{b_{t,i}^2}}\right)
>     \\\\
> &=\sum\limits_{k=0}^{t-1} \beta_1^{t-k} \|\| \nabla f(x_{k+1}) - \nabla f(x_k)\|\|\left(\sqrt{\sum\limits_{i=1}^d\frac{m_{k,i}^2}{b_{t,i}^2}}\right)
>     \\\\
> &\leq \sum\limits_{k=0}^{t-1} \beta_1^{t-k} L \|\| x_{k+1} - x_k\|\| \left(\sqrt{\sum\limits_{i=1}^d\frac{m_{k,i}^2}{b_{t,i}^2}}\right) \\\\
> &=  L\gamma\sum\limits_{k=0}^{t-1} \beta_1^{t-k} \left(\sqrt{\sum\limits_{i=1}^d\frac{m_{k,i}^2}{b_{k,i}^2}}\right)\left(\sqrt{\sum\limits_{i=1}^d\frac{m_{k,i}^2}{b_{t,i}^2}}\right) \\\\
> &\leq \frac{L\gamma}{c_m}\sum\limits_{k=0}^{t-1} \beta_1^{t-k} \sum\limits_{i=1}^d\frac{m_{k,i}^2}{b_{k,i}^2}.
> \end{align}
>
> Then, we plug this bound into the previous ones and get:
>
> $$f(x_{t+1}) - f(x_t) \leq -(1 - \beta_1)\gamma\sum\limits_{k=0}^t \beta_1^{t-k}\sum\limits_{i=1}^d\frac{\nabla_{k,i}g_{k,i}}{b_{t,i}} + \frac{L\gamma^2}{c_m}\sum\limits_{k=0}^{t} \beta_1^{t-k} \sum\limits_{i=1}^d\frac{m_{k,i}^2}{b_{k,i}^2}.$$
>
> Applying Lemma C.2 and summing over $t$, we derive the following bound:
>
> \begin{align}
> 	(f(x_T) - f^\ast) - (f(x_0) - f^\ast) &\leq -(1 - \beta_1)\gamma\sum\limits_{t=0}^{T-1}\sum\limits_{k=0}^t \beta_1^{t-k}\sum\limits_{i=1}^d\frac{\nabla_{k,i}g_{k,i}}{b_{t,i}} \\\\
> &+ \frac{L\gamma^2(1-\beta_1)}{c_m}\sum\limits_{t=0}^{T-1}\sum\limits_{k=0}^{t} \beta_1^{t-k} \sum\limits_{i=1}^d\sum\limits_{j=0}^k\beta_1^{k-j}\frac{g_{j,i}^2}{b_{k,i}^2}.
> \end{align}
>
> Then, similarly to the proof of Lemma C.3, we denote the coefficients in front of $\nabla_{r,i} g_{r,i}$ and $g_{r,i}^2$ as $-\gamma C_{r,i}$ and $A_{r,i}$ respectively. They have the following explicit formulas
>
> \begin{align}
>     -\gamma C_{r,i} &= -(1 - \beta_1)\gamma\sum\limits_{t=r}^{T-1}\frac{\beta_1^{t-r}}{b_{t,i}},\\
>     A_{r,i} &= \frac{L\gamma^2(1-\beta_1)}{c_m}\sum\limits_{t=r}^{T-1}\sum\limits_{k=r}^t\frac{\beta_1^{t - r}}{b_{k,i}^2}
> \end{align}
>
> After that, we follow exactly the same steps as in the proof of Theorem C.5 and get that for $\gamma \leq \frac{(1 - \beta_1)^2c_m^3 b_0}{2L}$ the following inequality holds:
>
> \begin{align}
> 	\sum\limits_{t=0}^{T-1}\sum\limits_{i=1}^d \frac{\gamma C_{t,i}}{2}\nabla_{t,i}^2 &\leq (f(x_0) - f^\ast) - (f(x_T) - f^\ast) - \sum\limits_{t=0}^{T-1}\sum\limits_{i=1}^d (\gamma C_{t,i} - 2A_{t,i})\nabla_{t,i}\theta^u_{t,i} \\\\
> &+ \sum\limits_{t=0}^{T-1} \sum\limits_{i=1}^d 2A_{t,i}(\theta^u_{t,i})^2 + \sum\limits_{t=0}^{T-1} \sum\limits_{i=1}^d \gamma C_{t,i}(\theta^b_{t,i})^2.
> \end{align}
>
> The rest of the proof follows very similar steps to the current proofs of Theorems C.5 and C.12 – from Theorem C.5 the same dependence on $1 - \beta_1$ follows, and from Theorem C.12 the dependence on $d$ is preserved.
>
> >2:**In Theorem 3.3, $M \to \Delta$**
>
> A: Thank you for spotting the typo: should be $M$ in the upper bound.

---

> > ### Comment · Reviewer_GE8p · 2025-04-01
> >
> > Thanks for the authors' feedback and I will keep my recommendation for acceptance.

---

> > > ### Author Response · Authors · 2025-04-08
> > >
> > > We thank the reviewer for checking our response and keeping the positive recommendation! We also thank the reviewer for the positive and encouraging feedback.
> > >
> > > If possible, we would be grateful if the reviewer could consider championing our paper during the discussion with other reviewers and the AC.

---

### Official Review · Reviewer_RQzs · 2025-03-12

**Overall Recommendation:** 1

**Summary:**

This paper provides a loss function that AdaGrad and Adam have bad high probability convergence when the noise is heavy-tailed. Then they show a desirable convergence rate for AdaGrad-Norm and Adam-Norm with clipping gradient to fix the heavy-tailed noise issue. They conduct experiments to show that clipping can help Adam optimize faster.

**Claims And Evidence:**

1. Theorem 2.1 can’t be viewed as an example of bad dependence on $\delta$. See detailed comments in theoretical claims.
2. Algorithm 1 is not a proper definition for Adam/AdaGrad. Instead, it should be Adam-Norm/AdaGrad-Norm because $b_t$ is the weighted average/sum of the square of gradient norm rather than entry-wise square of gradient.
3. There is a gap between the experiments and the theory. All the theoretical results are for Adam-Norm/AdaGrad-Norm but the experiments are done with Adam/AdaGrad.

**Essential References Not Discussed:**

[1] conducted experiments to show that heavy-tailed noise is not the major factor for the gap between Adam and SGD. They point out that Adam should be more similar to sign descent when handling heavy-tailed noise. I think the authors should discuss more whether their results on Adam-Norm can really suggest anything for Adam when dealing with heavy-tailed noise because Adam-Norm makes Adam completely different from sign descent.

References

[1] Kunstner, F., Chen, J., Lavington, J. W., and Schmidt, M. Noise is not the main factor behind the gap between SGD and Adam on Transformers, but sign descent might be. arXiv preprint arXiv:2304.13960.

**Experimental Designs Or Analyses:**

1. The noise in the quadratic experiment seems too large. I feel the failure of Adam comes from the large magnitude of noise rather than heavy tail. If large gradients are accumulated in $b_t$ without clipping, the effective stepsize of AdaGrad will be much smaller than the effective stepsize of Clip-AdaGrad, which may be the real reason of AdaGrad’s failure. I’d like to see how the result will change if you use 100 times smaller noise. The distribution is still heavy-tailed. Or a fair comparison between AdaGrad and Clip-AdaGrad is to thoroughly tune the learning rate in a large range for each one. Comparing them under the same learning rate rather than the optimal learning rate is not appropriate.
2. What is the randomness in the fine-tuning experiment? Is it just the seed that determines the batch order in training set? It is hard to believe that some seeds will lead to no improvement to val loss.

**Methods And Evaluation Criteria:**

High-probability convergence is preferable to in-expectation convergence, and heavy-tailed noise is a real issue in practice, as confirmed by this paper and prior work. While the fine-tuning experiment is reasonable, I am curious why the authors chose ALBERT over BERT and RoBERTa. Additionally, CoLA and RTE are relatively small datasets, though I don’t doubt that clipping would also be beneficial for larger datasets, given its common use in practice. However, it would be more informative to compare training loss rather than validation loss, as the focus is on optimization rather than generalization.

**Other Comments Or Suggestions:**

The introduction should be carefully rewritten since the authors often confuse Adam with Adam-Norm. It seems that the authors initially focus on Adam but only get rigorous theoretical results for Adam-Norm so they need to make the current title. But the introduction and the paper should be revised to keep consistency.

**Other Strengths And Weaknesses:**

Weaknesses:
1. It is hard to understand whether the authors really want to analyze clipping for Adam or Adam-Norm. The title and the theoretical results are all about Adam-Norm. But the motivation for analyzing Adam-Norm under heavy-tailed noise is missing because the relevant literature mentioned in introduction focuses on Adam under heavy-tailed noise. If the goal is to analyze Adam through the middle step of Adam-Norm, then it is doubted how much insight can be provided by the theoretical results because Adam and Adam-Norm are completely different algorithms as shown in [2].
2. There is no clear definition of heavy-tailed noise. The failure example provided in theorem 2.1 is not heavy-tailed noise in my opinion.

Reference

[2] Xie, S., Mohamadi, M. A., and Li, Z. Adam exploits $\ell_\infty$-geometry of loss landscape via coordinate-wise adaptivity. arXiv preprint arXiv:2410.08198.

**Questions For Authors:**

1. I’d like to see some discussion on Theorem 2.1 to check if I misunderstand anything. See details in theoretical claims.
2. Why can we view Theorems 3.1–3.3 as a positive result? Even if the number of iterations doesn’t depend on $\delta$, its dependence on $\epsilon$ is even worse than Theorem 2.1 if Theorem 2.1 is valid.
3. I don’t understand why the hyperparameters need to depend on $\delta$ in Theorems 3.1–3.3. If we want very high probability $1-\delta$, then $A$ needs to be very large, and $\gamma$ and $\lambda$ will be very small. Does this suggest that the clipping effect needs to be very strong to achieve a good convergence result?
4. Can you conduct more experiments as suggested in the experimental design?
    a. Decrease the magnitude of noise in the quadratic problem and carefully tune learning rate.
    b. Report the training loss of the fine-tuning experiment.
    c. Repeat the fine-tuning experiments with Clip-Adam-Norm.

**Relation To Broader Scientific Literature:**

It can help understand the relationship between vanilla optimization algorithms and the tricks like clipping that people use in practice.

**Theoretical Claims:**

I checked the proof of theorem 2.1 and found the statement and loss example problematic. I have listed the specific problems below but in short I find this theorem completely useless. We can easily get similar result even for deterministic Adam because the key idea in the loss example is to make $x_0$ far enough from $x^*$ so that it can not enter the region with small loss because its moving distance is restricted when $T$ and stepsize $\gamma$ are not large enough.
1. This **cannot** be viewed a polynomial dependence on $\delta^{-1/2}$ and $\epsilon^{-1/2}$ because both the function and noise depends fixed $\epsilon$ and $\delta$. They are defined by the specific $\epsilon$ and $\delta$ and then the authors show that $T$ needs to be larger than $\frac{R}{\gamma} (\frac{\sigma}{\nu \sqrt{2 \delta}}-1)$. For each fixed function, you instead need to show the convergence rate as a function of $\epsilon$ and $\delta$.
2. It is inapproriate to choose initial $x_0$ depending on stepsize $\gamma$. Specifically, clip-Adam will get the same negative result under the same setting so theorem 2.1 isn't a valid example for failure of Adam without clipping. In comparison, stepsize is decided based on the lipschitzness of loss function and initial distance $R=||x_0-x^*||$ in the positive result theorem 3.1-3.3.
3. The noise distribution is not heavy-tailed at all since there is only noise at the first update and the noise even has bounded variance. It is very different from the heavy-tailed noise people find in practice.
4. The provided example doesn’t satisfy assumption 1.1 as claimed in the theorem statement. The variance of noise is always $1$ rather than any choice of $\sigma$.

---

> ### Author Rebuttal · Authors · 2025-04-01
>
> We thank the reviewer for the valuable feedback that helped improve our paper. The requested numerical experiments can be found here: https://anonymous.4open.science/r/Clip-Adam-Norm-and-Clip-AdaGrad-Norm-1E8A/ (folder "rebuttal").
>
> >1: **Algorithm 1.**
>
> A: We will add "-norm" to method names in Algorithm 1. Section 1.3 provides coordinate-wise versions of AdaGrad/Adam.
>
> >2:**Gap between experiments and theory.**
>
> A: We include coordinate-wise methods with $\beta_1 = 0$ in Appendix C.5 (end of Section 3). **Our proofs now cover coordinate-wise stepsizes—see our response to Reviewer GE8p**. In Section 2, our results hold in 1D, where coordinate-wise and norm/scalar versions coincide. Hence, the theory aligns well with quadratic problem experiments. The Clip-Adam experiment without delay, though not analyzed theoretically, complements our theory.
>
> >3:**Why ALBERT?**
>
> A: Mosbach et al. (2020) motivate our choice. RoBERTa wasn't used with gradient clipping in their experiments. Due to computational constraints and the need for 100 runs per method for high-probability convergence, we used a smaller model (ALBERT) instead of BERT.
>
> >4:**Small datasets.**
>
> A: We use two of three datasets from Mosbach et al. (2020). We can add results on larger datasets if the paper gets accepted.
>
> >5: **Training loss.**
>
> A: New plots are available in the anonymized repository.
>
> >6: **On Theorem 2.1.**
>
> A: Let us address your concerns on the example of AdaGrad with momentum (Theorem B.4). The same reasoning holds for other methods as well.
>
> 1. We focus on worst-case guarantees [1], estimating complexity for the worst possible problem given the method’s parameters. This allows a resisting oracle [2], where the problem depends on $\varepsilon$ and $\delta$. This is a classical approach.
>
> 2. We require $x_0 > \sqrt{2\varepsilon} + 3\gamma$. Since typically $\varepsilon, \gamma \ll 1$, this assumption is mild. The stepsizes in Theorems 3.1-3.3 do not contradict Theorem 2.1: the results are derived for *different methods*. For $R, \sigma, M, L, b_{-1} = O(1)$ and large $K$ in Theorem 3.3, Theorem B.4 applies.
>
> 3. The fact that AdaGrad/Adam lack logarithmic dependence on $1/\delta$ even for $\alpha = 2$ strengthens our results. Some works [3] label $\alpha = 2$ case as heavy-tailed. Our setting, where noise is zero except in the first iteration, reinforces our findings: a large first stochastic gradient drastically reduces stepsize.
>
> 4. The stochastic gradient noise is $\sigma \xi_k$, with $\mathbb{E}[\xi_0^2] = 1 \Longrightarrow \mathbb{E}[\sigma^2\xi_0^2] = \sigma^2$.
>
> >7: **The noise in the quadratic experiment seems too large.**
>
> A: Figures 1, 3, and 4 show results for different $\gamma$. Methods with clipping consistently outperform those without. Additional experiments with lower noise (scaled by 100) and tuned stepsizes are in the anonymized repository.
>
> >8: **Fine-tuning experiment?**
>
> A: Randomness stems from the seed that determines mini-batch sampling order.
>
> >9: **Results from Kunstner et al. (2023)**
>
> A: We will discuss this paper in the final version. Their work shows Adam outperforming SGD even in full-batch settings -- this observation is unrelated to stochasticity. Our focus is on high-probability convergence of AdaGrad/Adam-based methods, showing that without clipping, they have poor high-probability complexities, similar to SGD. Thus, our results complement Kunstner et al. (2023) by highlighting the necessity of gradient clipping for high-probability convergence.
>
> >10:**Scalar or coordinate-wise methods?**
>
> A: We analyze both scalar (norm) and coordinate-wise versions of AdaGrad/Adam, studying both in experiments.
>
> >11:**Introduction.**
>
> A: We will rewrite the introduction for clarity on considered methods.
>
> >12:**Definition of heavy-tailed noise.**
>
> A: The noise is heavy-tailed if it satisfies Assumption 1.1.
>
> >13:**Discussion of Theorem 2.1.**
>
> A: We will expand the discussion, incorporating the points above.
>
> >14: **Dependence of $K$ on $\varepsilon$ and $\delta$ in Theorems 3.1-3.3.**
>
> A: The complexity results are presented in a standard way for the optimization literature (e.g., see Theorem 2.4 from [4]). The hyperparameters also depend on $\varepsilon$ and $\delta$: even standard SGD with constant stepsize converges only to the neighborhood of the solution for strongly convex smooth problems, i.e., one has to choose the stepsize to be dependent on the target error. We also provide the rates in Appendix C (see Theorems C.5, C.7, C.9, C.12).
>
> ---
>
> References:
>
> [1] A. Nemirovsky & D. Yudin. Problem complexity and model efficiency in optimization. J. Wiley @ Sons, New York (1983)
>
> [2] Y. Nesterov. Lectures on convex optimization. Springer (2018)
>
> [3] E. Gorbunov et al. Stochastic optimization with heavy-tailed noise via accelerated gradient clipping. NeurIPS (2020)
>
> [4] S. Ghadimi & G. Lan. Stochastic first-and zeroth-order methods for nonconvex stochastic programming. SIAM journal on optimization (2013)

---

> > ### Comment · Reviewer_RQzs · 2025-04-02
> >
> > Thanks for your clarifications. Some of my concerns are addressed. But I still have some questions. Most importantly, I can’t raise my score since I still find theorem 2.1 problematic.
> >
> > I would like to restate my concern on theorem 2.1 because the rebuttal doesn’t address it. Thanks for your explanation and now I agree that we can choose loss function depending on $\epsilon$ and $\delta$. However, I think the bigger issue is that you choose the initialization based on fixed stepsize. You argue that $x_0 > \sqrt{2 \epsilon} + 3 \gamma$ is a mild assumption because $\gamma$ is small. Even if $\gamma$ is small, the fact that $x_0$  must be set as a function of $\gamma$ is the core issue. A proper lower bound or negative example should demonstrate that the algorithm fails on a fixed problem — not just that for each $\gamma$, there exists some adversarial problem where it fails. A fixed step size of course shouldn’t work for any problem. You also mention this in your discussion of theorem 3.1-3.3. For the result to reflect a true limitation of M-AdaGrad, there should exist a fixed function and initialization $x_0$  such that M-AdaGrad fails regardless of $\gamma$. That is not demonstrated here.
> >
> > I have checked the plots for rescaling noise. AdaGrad and Adam can reach a relative low loss very quickly while clip-AdaGrad and clip-Adam optimizes very slowly in the beginning. The only advantage of clip version is that they can reach a slightly lower loss than those without clipping after several thousand steps. So I don’t think this is a valid experiment for supporting the claim that clipping can improve optimization under noise. Optimization actually becomes more slowly with clipping operation.

---

> > > ### Author Response · Authors · 2025-04-04
> > >
> > > Dear Reviewer RQzs,
> > >
> > > Thank you for your further comments.
> > >
> > > ## On Theorem 2.1
> > >
> > > We would like to further clarify the construction used in Theorem 2.1 ( and in Theorems B.4, B.8, B.12, and B.16 from the Appendix, respectively) and the interpretation of the results.
> > >
> > > 1. We highlight that Theorem 2.1 is not a classical *lower bound*, but rather a *negative result*. The parameters of the problem and the method (e.g., M-AdaGrad) are: $\varepsilon$ – target optimization error, $\delta$ – failure probability, $b_{-1}$ or $b_{0}$ – initial scaling factor, $||x_0 - x^\ast||$ -- initial distance to the optimum, $\gamma$ -- stepsize. The goal of Theorem 2.1 is not to show that for any set of the above parameters, the number of iterations required for achieving $\varepsilon$-solution with probability $\geq 1-\delta$ is proportional to $\frac{1}{\delta^c}$ for some $c > 0$, but to show the same for some (quite noticeable) range of parameters. We emphasize that $x_0$ in our example is not a function of $\gamma$: our result holds whenever $x_0 > \sqrt{2\varepsilon} + 3\gamma$, which is not equivalent to assuming that $x_0$ is a function of $\gamma$. Moreover, this restriction is typically satisfied in practice since typically $||x_0 - x^\ast|| \geq 1$, $\varepsilon$ is small (not greater than $10^{-3}$) and $\gamma$ is small (e.g., the default value of $\gamma$ in PyTorch is $0.01$). Therefore, Theorem 2.1 is a meaningful negative result with practical implications. Nevertheless, you are right that it is not a lower bound since our construction does not hold for any choice of parameters. It leads us to the natural question: Is Theorem 2.1 sufficient to claim that adaptive schemes like AdaGrad and Adam have bad high-probability convergence? To answer this question, let us assume the opposite.
> > >
> > > 2. Assume that M-AdaGrad (or Adam) does converge for some range of $\gamma$ with high-probability under the settings of Theorem 2.1 and assume that the complexity depends on $\delta$ through the factor of $\mathrm{poly}(\log^{c_1}(1/\delta))$ for some $c_1 > 0$, i.e., the complexity of finding $\varepsilon$-solution after $K$ steps with probability $\geq 1-\delta$ (under convexity, smoothness and bounded variance assumption) is
> > > $$K = \mathcal{O}\left(\frac{1}{ \mathrm{poly}(\gamma^{c_2}) \mathrm{poly}(\varepsilon^{c_3})} \mathrm{poly}(\log^{c_2}\frac{1}{\delta})\right)$$
> > > for some $c_1, c_2, c_3 \geq 0$ and under some conditions on $\gamma$. We are not aware of any results in the literature stating the convergence of AdaGrad/Adam-type methods only for large enough $\gamma$. In contrast, the existing results for these methods hold either for all $\gamma > 0$ or for any (e.g., Theorem 1 from [1]) $0 < \gamma < C$ for some $C$ (e.g., Case 1 of Theorem 1 from [2]). However, the aforementioned hypothetical result for AdaGrad/Adam cannot hold for any $0 < \gamma < C$ regardless of $C$ since it would contradict Theorem 2.1. Therefore, we conjecture that the result of Theorem 2.1 can be extended to any choice of $\gamma$. We will explicitly mention it in the final version.
> > >
> > > Overall, the above arguments show that Theorem 2.1 is a quite strong negative result for AdaGrad/Adam without clipping. However, we promise to add more discussion of this result in the final version.
> > >
> > > ---
> > > References
> > >
> > > [1] Défossez et al. A simple convergence proof of Adam and Adagrad, TMLR 2022
> > >
> > >
> > > [2] Xie et al. Linear convergence of adaptive stochastic gradient descent, ICML 2020
> > >
> > >
> > > ---
> > >
> > > ## Experiments
> > >
> > > In the provided experiments, the goal was to illustrate that the methods with clipping achieve better optimization error with higher probability. Indeed, the plots are given in logarithmic scale in the Y-axis, but “the width of the oscillations” is of the same size on the plots. This means that the methods that achieve better error also oscillate less. We also uploaded additional plots to the same file with illustrations of methods’ behavior with different stepsizes (see the very end of “description_of_the_results_and_plots.md”). These plots indicate that with very small stepsizes AdaGrad and Adam do not reach a reasonable error. Moreover, the small reduction of the stepsizes does not significantly improve the achieved optimization error.
> > >
> > > We also highlight that the contribution of our paper is primarily theoretical.
> > >
> > > ---
> > >
> > > **If you have any further comments or questions, we kindly ask you to let us know: we are committed to promptly addressing any remaining concerns.**
> > >
> > > Best regards,
> > >
> > > Authors

---

### Official Review · Reviewer_9uKJ · 2025-03-12

**Overall Recommendation:** 3

**Summary:**

This work examines adaptive optimization methods, specifically variants of Adam and AdaGrad, in settings with heavy-tailed noise. The authors establish that for Adam and AdaGrad with momentum, achieving a polylogarithmic dependence on the confidence level is impossible. In contrast, they show that the clipped versions of Adam-Norm and AdaGrad-Norm provide high-probability convergence guarantees, with additional results for the clipped versions of Adam and AdaGrad without momentum. Numerical experiments further confirm the improved performance of the clipped variants.

## update after rebuttal

I thank the authors for their response and I updated my score.

I recommend the authors to better reflect the paper is mainly focused on the norm versions instead of the element-wise versions per the comments of the other reviewers regarding Algorithm 1.

**Claims And Evidence:**

The paper provides an analysis of several variants of Adam and AdaGrad to support the claim that clipping improves performance in the presence of heavy-tailed noise. However, some aspects of the evidence could be more comprehensive.

First, the noise model deviates from the standard gradient oracle model, as the noise may depend on time. This is only briefly mentioned (in line 90 of the second column) and is not explicitly detailed in Assumption 1.1, despite being a crucial component for establishing the lower bound (see the noise distribution in line 928).

Additionally, while AdaGrad and Adam perform poorly without clipping, the results for clipped coordinate-wise methods remain incomplete, as momentum is not incorporated. So, claiming in the abstract that clipping fixes Adam is not fully proven.

Finally, Theorem 3.3 is established using Assumption 1.4, whereas the lower bound does not support this assumption.

**Essential References Not Discussed:**

Essential references are discussed.

**Experimental Designs Or Analyses:**

The experiments align with the authors' claims. Notably, Adam appears to fail completely with a reasonable probability in the CoLa experiment (Figure 2), which is somewhat surprising. The reviewer would expect that, with properly tuned parameters, Adam would still be able to learn, even if the clipped version performs better.

**Methods And Evaluation Criteria:**

Yes

**Other Comments Or Suggestions:**

No further comments.

**Other Strengths And Weaknesses:**

**Strengths**
- The analysis of widely used optimization methods under heavy-tailed noise is an important topic.
- This paper establishes results for a broad range of adaptive variants.

**Weaknesses**
- As mentioned earlier, the connection between scalar and coordinate-wise versions is limited.
- The large number of details, different variants, and assumptions make it challenging for readers to compare the positive and negative results presented in the work.

**Questions For Authors:**

- Can the lower bounds be modified to accommodate stochastic gradients that does not depend on the time? What modifications are required and how difficult is the task?
- Can an improved lower bound support assumption 1.4?
- What is the difficulty in removing assumption 1.4 from Theorem 3.3?
- What is the main difficulty in accommodating both coordinate-wise algorithms and momentum when discussing heavy-tailed noise?
- With light-tailed noise, AdaGrad-Norm converge even without accurate specification of the hyper-parameters (the bound itself degrades according to the hyper-parameters, but the algorithm still converge). Can such results be obtained with heavy-tailed noise?

Overall, the paper provides new insights and results, but the connection between claims and results can be improved.

**Relation To Broader Scientific Literature:**

The paper aims to extend observations from Clip-SGD to adaptive methods, which is a valuable direction for advancing our understanding of optimization with heavy-tailed noise.

**Theoretical Claims:**

The paper contains many technical details, most of which are provided in the appendix. The reviewer focused on the lower bound of M-AdaGrad and the upper bound of Clip-M-AdaGradD/Clip-AdamD-Norm presented there.

Aside from the non-standard noise model, which the reviewer noted is used in the appendix, the examined parts appear to be correct.

---

> ### Author Rebuttal · Authors · 2025-04-01
>
> We thank the reviewer for the constructive feedback that helped to improve our paper.
>
> >1:**Time-dependent noise.**
>
> A: Thank you for spotting this. We will rewrite Assumption 1.1 to explicitly highlight that we allow time-dependent noise. Such assumptions are relatively standard for stochastic optimization with streaming oracle [1-3]. Moreover, many existing convergence upper bounds hold for time-varying noise as long as the corresponding moment bound is satisfied for each step [4-6]. We will add these clarifications to the final version of our paper.
>
> >2:**Coordinate-wise methods & momentum.**
>
> A: In the abstract, we claim that we show that clipping fixes AdaGrad-Norm and Adam-Norm. In Section 1.1, we made a typo in line 65: it should be “without momentum” instead of “with momentum”. We also explicitly mention that Clip-AdaGradD and Clip-AdamD are analyzed without momentum. **To address this limitation of our paper, we generalized our proofs for the methods with coordinate-wise stepsizes to the case of $\beta_1 > 0$ – please see our response to Reviewer GE8p.** We promise to add the complete proof to the final version of our paper.
>
> >3:**Lower bound & Assumption 1.4.**
>
> A: Our lower bounds can be extended to support Assumption 1.4, if we modify $f$ as follows:
>
> $$f(x) = \begin{cases}\frac{1}{2}x^2,& \text{if } |x| < \nu,\\\\ \nu\left(|x| - \frac{1}{2}\nu\right),& \text{if } \nu \leq  |x| \leq D,\\ \nu\left(D - \frac{1}{2}\nu\right),& \text{if } |x| > D, \end{cases}$$
>
> where $D > |x_0| > \nu$. Then, the proofs remain the same, and the problem satisfies Assumption 1.4.
>
> >4:**Lower bound & time-independent noise.**
>
> A: If we assume that the noise is i.i.d. for each step, e.g., the same as for $k = 0$ in formula (16), then for the provided example of function and oracle M-AdaGrad/Adam should also have inverse-power dependence on $\delta$ in their high-probability complextites – this is our conjecture. However, deriving the lower bounds for this case is technically more difficult because the explicit form of the iterates becomes more involved than in Lemma B.1 (due to the summation of the squared norms of stochastic gradients in the stepsize). We will discuss this more explicitly in the final version of the paper.
>
> >5: **Why Assumption 1.4?**
>
> A: The main difficulty comes from the dependence of $b_t$ and $g_t$ in the methods without delayed stepsizes. The existing approaches typically use boundedness of the variance and the norm of the gradient (see Lemma 5.1 in [7]) or assume that the noise is sub-Gaussian [8] to tackle this issue. In the heavy-tailed noise regime, these assumptions do not hold. Therefore, we use Assumption 1.4, which is a relaxation of the assumption used in [9] (see also our discussion of Theorem 3.3 on page 7). More precisely, to decouple $b_t$ and $g_t$ in the analysis, we multiply the inequality above (67) by $b_t$. However, it eventually leads to the non-trivial weighted sum of function values: the first sum in the RHS of the first row on page 44. After small rearrangements, we get the term $\sum_{t=1}^T \left( \frac{b_t}{p_t} - \frac{b_{t-1}}{p_{t-1}} \right)(f(x_t) - f_\ast)$ that we estimate using Assumption 1.4. We are not aware of the alternative ways of analyzing versions of AdaGrad/Adam or closely related methods in the heavy-tailed noise regime.
>
> >6:**On the hyper-parameters**
>
> A: For AdaGrad-Norm *without clipping*, similar results cannot be obtained with logarithmic dependence on $1/\delta$ due to Theorem 2.1. However, it is an interesting open question whether it is possible to show convergence for Clip-AdaGrad-Norm with a choice of hyper-parameters agnostic to $L,\sigma, \alpha$, and $R$, in the heavy-tailed noise regime. We leave this question for future work.
>
> ---
>
> References:
>
> [1] S. Ghadimi & G. Lan. Optimal Stochastic Approximation Algorithms for Strongly Convex Stochastic Composite Optimization, II: Shrinking Procedures and Optimal Algorithms. SIOPT 2012
>
> [2] N. J. A. Harvey et al. Simple and optimal high-probability bounds for strongly-convex stochastic gradient descent. arXiv:1909.00843
>
> [3] A. Sadiev et al. High-probability bounds for stochastic optimization and variational inequalities: the case of unbounded variance. ICML 2023
>
> [4] J. Zhang et al. Why are Adaptive Methods Good for Attention Models? NeurIPS 2020
>
> [5] A. Cutcosky & H. Mehta. High-probability bounds for Non-Convex Stochastic Optimization with Heavy Tails. NeurIPS 2021
>
> [6] T. D. Ngyen et al. Improved convergence in high probability of clipped gradient methods with heavy tailed noise. NeurIPS 2023
>
> [7] A. Défossez et al. A Simple Convergence Proof of Adam and Adagrad. TMLR 2022
>
> [8] Z. Liu et al. High Probability Convergence of Stochastic Gradient Methods. ICML 2023
>
> [9] S. Li & Y. Liu. High probability analysis for non-convex stochastic optimization with clipping. ECAI 2023

---

> > ### Comment · Reviewer_9uKJ · 2025-04-08
> >
> > I thank the authors for their response and I updated my score.
> >
> > I recommend the authors to better reflect the paper is mainly focused on the norm versions instead of the element-wise versions per the comments of the other reviewers regarding Algorithm 1.
> >
> > I do not have further questions.

---

> > > ### Author Response · Authors · 2025-04-08
> > >
> > > We thank the reviewer for checking our replies and updating the score. We promise to better reflect in the introduction that most of the results are derived for the scalar versions of AdaGrad and Adam with clipping, and we will apply necessary corrections according to the reviews. We will also extend our analysis of coordinate-wise versions of AdaGradD and AdamD with clipping to the case of $\beta_1 > 0$, following the arguments provided in our response to Reviewer GE8p.
> > >
> > > Thank you once again for your insightful feedback.

---

### Official Review · Reviewer_H5h1 · 2025-03-12

**Overall Recommendation:** 3

**Summary:**

This paper suggests that clipping enables high-probability convergence (with polylogarithmic dependence on the confidence level δ) for Adam-norm/AdaGrad-norm under heavy-tailed noise. In contrast, without clipping, Adam/AdaGrad has inverse-power dependence.
The authors provide some numerical results to support this result.

**Claims And Evidence:**

The claim of this work is mainly supported by providing the theoretical results and comparing it to the settings / results in previous works.
Overall the paper is written clearly to deliver the ideas.

**Essential References Not Discussed:**

As far as I’m concerned, the paper seems to address previous / concurrent related works well enough.

**Experimental Designs Or Analyses:**

It’s not the specific settings but rather the scale that I’m concerned with. I believe the experiments can be much more comprehensive to claim a solid support of the theory results.

**Methods And Evaluation Criteria:**

The authors attempt to validate the idea on both synthetic and realistic problems in numerical experiments.

**Other Comments Or Suggestions:**

none

**Other Strengths And Weaknesses:**

* The obvious weakness of this work is that the versions of Adam/AdaGrad are actually normed ones, i.e., Adam-norm/AdaGrad-norm, rather than the original form. This is a concern on both theory and practice sides. I think this should be discussed more clearly / up front in the paper rather than at the end of the theory result.
* I kind of received an impression that various versions of algorithms are analyzed not because they are all important but because they can be proved. It would be nice to see how the choice of conditions under which methods are analyzed is made, or why analyzing all of these would be worthwhile.
* Numerical results for CLIP-Adam do not seem strong or much differentiated.

**Questions For Authors:**

none

**Relation To Broader Scientific Literature:**

This work renders a theoretical contribution to the optimization for machine learning community in particular for adaptive methods and more broadly stochastic methods. The authors present a balanced review over prior works in the Related Work section and the New Upperbound section, where one can find how this work compares to previous works.

**Theoretical Claims:**

I’ve skimmed through proofs of failure cases in Appendix B. I’ve not seen the proofs for the clipping results.

---

> ### Author Rebuttal · Authors · 2025-04-01
>
> Thank you for your time and positive feedback. We also thank you for your useful comments, which we have addressed below.
>
> >1:**It’s not the specific settings but rather the scale that I’m concerned with. I believe the experiments can be much more comprehensive to claim a solid support of the theory results.**
>
> A: We agree that the scale of our experiments can be further improved, and we can provide additional numerical results for larger models and datasets. However, the main contribution of our paper is primarily theoretical. Therefore, we see our experimental results as complementary contributions extending theoretical findings.
>
> >2: **The obvious weakness of this work is that the versions of Adam/AdaGrad are actually normed…**
>
> A: We point out that we also analyze coordinate-wise versions of AdaGrad and Adam (with $\beta_1 = 0$) in Appendix C.4, as explained in the very end of Section 3. Moreover, we generalized our proof to the case of $\beta_1 > 0$ (please, see our response to Reviewer GE8p).  Unfortunately, due to the page limitations of the main part, we had to defer these results to the Appendix. If the paper gets accepted, we will use an additional page in the main part to discuss the results on coordinate-wise versions of AdaGrad and Adam with clipping in more detail.  We also kindly ask you to check our response to Reviewer RQzs (comments 2 and 10).
>
> >3: **I kind of received an impression that various versions of algorithms are analyzed not because they are all important but because they can be proved. It would be nice to see how the choice of conditions under which methods are analyzed is made, or why analyzing all of these would be worthwhile.**
>
> A: Thank you for mentioning this point of view. The main goal of this paper is to provide a comprehensive answer to the questions formulated on the second page, right before Section 1.1.  In general, the answer could potentially depend on the choice of the convexity and the stepsizes. Therefore, we analyze both non-convex and convex problems and derive the results for the scalar (a.k.a. “-norm”) and coordinate-wise versions of AdaGrad and Adam with and without delayed stepsizes. We believe that the provided analysis is quite comprehensive and covers many different cases (though not all possible ones for the sake of having the paper of a reasonable length; see also our response to Reviewer RQzs, comment 10). That is, we believe that the main research questions formulated in the paper are adequately addressed, i.e., we show that standard AdaGrad and Adam (with and without delayed stepsizes) do not enjoy logarithmic dependence on $1/\delta$ in their high-probability complexities and clipping provably improves their high-probability convergence.
>
> >4:**Numerical results for CLIP-Adam do not seem strong or much differentiated.**
>
> A: We believe that the main theoretical findings of this paper are properly illustrated and complemented by our numerical results. We kindly ask the reviewer to clarify the concern.

---

### Decision · Program_Chairs · 2025-05-01

**Decision:**

Accept (poster)

**Comment:**

The paper studies the convergence under heavy tail noise of adaptive methods such as the single step size versions of Adagrad and Adam and other variants of these methods. The paper gives an example of an instance where the iteration complexity of these methods is polynomially large in $1/\delta$, where $\delta$ is the failure probability. The paper then shows that clipping improves the convergence, and establishes convergence guarantees with a logarithmic dependence on $1/\delta$ for the methods with clipping.

Adaptive methods such as AdaGrad and Adam are some of the widely used optimization methods, and understanding their performance in challenging stochastic settings is an impactful direction. The paper makes a valuable contribution in this direction and further underscores the importance of clipping in settings where the stochastic noise is heavy-tailed. Most of the reviewers appreciated the main contribution of this work and supported accepting the paper. One of the reviewers was concerned that Thm 2.1 does not provide a lower bound but rather only an example of an instance, and that the experimental evaluation has limitations. Although Thm 2.1 does not provide an actual lower bound, it provides an informative example. We encourage the authors to strengthen the experimental evaluation based on the reviewers' feedback. Overall, the paper makes a valuable theoretical contribution that is sufficiently supported by the experimental evaluation.